# LATENT BOTTLENECKED ATTENTIVE NEURAL PROCESSES

**Leo Feng**
Mila – Université de Montréal
& Borealis AI
`leo.feng@mila.quebec`

**Hossein Hajimirsadeghi**
Borealis AI
`hossein.hajimirsadeghi@borealisai.com`

**Yoshua Bengio**
Mila – Université de Montréal
`yoshua.bengio@mila.quebec`

**Mohamed Osama Ahmed**
Borealis AI
`mohamed.o.ahmed@borealisai.com`

## ABSTRACT

Neural Processes (NPs) are popular methods in meta-learning that can estimate predictive uncertainty on target datapoints by conditioning on a context dataset. Previous state-of-the-art method Transformer Neural Processes (TNPs) achieve strong performance but require quadratic computation with respect to the number of context datapoints, significantly limiting its scalability. Conversely, existing sub-quadratic NP variants perform significantly worse than that of TNPs. Tackling this issue, we propose Latent Bottlenecked Attentive Neural Processes (LBANPs), a new computationally efficient sub-quadratic NP variant, that has a querying computational complexity *independent* of the number of context datapoints. The model encodes the context dataset into a constant number of latent vectors on which self-attention is performed. When making predictions, the model retrieves higher-order information from the context dataset via multiple cross-attention mechanisms on the latent vectors. We empirically show that LBANPs achieve results competitive with the state-of-the-art on meta-regression, image completion, and contextual multi-armed bandits. We demonstrate that LBANPs can trade-off the computational cost and performance according to the number of latent vectors. Finally, we show LBANPs can scale beyond existing attention-based NP variants to larger dataset settings.

## 1 INTRODUCTION

Meta-learning aims to learn a model that can adapt quickly and computationally efficiently to new tasks. Neural Processes (NPs) are a popular method in meta-learning that models a conditional distribution of the prediction of a target datapoint given a set of labelled (context) datapoints, providing uncertainty estimates. NP variants (Garnelo et al., 2018a; Gordon et al., 2019; Kim et al., 2019) adapt via a conditioning step in which they compute embeddings representative of the context dataset. NPs can be divided into two categories: (1) computationally efficient (sub-quadratic complexity) but poor performance and (2) computationally expensive (quadratic complexity) but good performance. Early NP variants were especially computationally efficient, requiring only linear computation in the number of context datapoints but suffered from underfitting and as a result, overall poor performance. In contrast, recent state-of-the-art methods have proposed to use self-attention mechanisms such as transformers. However, these state-of-the-art methods are computationally expensive in that they require quadratic computation in the number of context datapoints. The quadratic computation makes the method inapplicable in settings with large number of datapoints and in low-resource settings. ConvCNPs (Gordon et al., 2019) partly address this problem by proposing to use convolutional neural networks to encode the context dataset instead of a self-attention mechanism, but this (1) requires the data to satisfy a grid-like structure, limiting the method to low-dimensional settings, and (2) the recent attention-based method Transformer Neural Processes (TNPs) have been shown to greatly outperform ConvCNPs.

| Method | Computational Complexity (Big-O) | | |
| | Training | Evaluation | |
| | Step | Condition | Query |
|---|---|---|---|
| CNP (Garnelo et al., 2018a) | $N + M$ | $N$ | $M$ |
| CANP (Kim et al., 2019) | $N^2 + NM$ | $N^2$ | $NM$ |
| NP (Garnelo et al., 2018b) | $N + M$ | $N$ | $M$ |
| ANP (Kim et al., 2019) | $N^2 + NM$ | $N^2$ | $NM$ |
| BNP (Lee et al., 2020) | $(N + M)K$ | $KN$ | $KM$ |
| BANP (Lee et al., 2020) | $(N^2 + NM)K$ | $KN^2$ | $KNM$ |
| TNP-D (Nguyen & Grover, 2022) | $(N + M)^2$ | — | $(N + M)^2$ |
| **EQTNP** (Ours) | $N^2 + NM$ | $N^2$ | $NM$ |
| **LBANP** (Ours) | $(N + M + L)L$ | $NL + L^2$ | $ML$ |

Table 1: Computational Complexity in Big-O notation of the model with respect to the number of context datapoints ($N$) and the number of target datapoints per batch ($M$). $L$ and $K$ are prespecified hyperparameters. $L$ is the number of latent vectors. $K$ is the number of bootstrapping samples for BNP and BANP. It is important that the cost of performing the query step is low.

Inspired by recent developments in efficient attention mechanisms, (1) we propose Latent Bottlenecked Attentive Neural Processes (LBANP), a computationally efficient NP variant that has a querying computational complexity *independent* of the number of context datapoints. Furthermore, the model requires only linear computation overhead for conditioning on the context dataset. The model encodes the context dataset into a fixed number of latent vectors on which self-attention is performed. When making predictions, the model retrieves higher-order information from the context dataset via multiple cross-attention mechanisms on the latent vectors. (2) We empirically show that LBANPs achieve results competitive with the state-of-the-art on meta-regression, image completion, and contextual multi-armed bandits. (3) We demonstrate that LBANPs can trade-off the computational cost and performance according to the number of latent vectors. (4) In addition, we show that LBANPs can scale to larger dataset settings where existing attention-based NP variants fail to run because of their expensive computational requirements. (5) Lastly, we show that similarly to TNPs, we can propose different variants of LBANPs for different settings.

## 2 BACKGROUND

### 2.1 META-LEARNING FOR PREDICTIVE UNCERTAINTY ESTIMATION

In meta-learning, models are trained on a distribution of tasks $\Omega(\mathcal{T})$. During each meta-training iteration, a batch of $B$ tasks $\mathbf{T} = \{\mathcal{T}_i\}_{i=1}^{B}$ is sampled from a task distribution $\Omega(\mathcal{T})$. A task $\mathcal{T}_i$ is a tuple $(\mathcal{X}, \mathcal{Y}, \mathcal{L}, q)$, where $\mathcal{X}$ is the input space, $\mathcal{Y}$ is the output space, $\mathcal{L}$ is the task-specific loss function, and $q(x, y)$ is a distribution over data points. During each meta-training iteration, for each $\mathcal{T}_i \in \mathbf{T}$, we sample from $q_{\mathcal{T}_i}$: a context dataset $\mathcal{D}_i^{\text{context}} = \{(x, y)^{i,j}\}_{j=1}^{N}$ and a target dataset $\mathcal{D}_i^{\text{target}} = \{(x, y)^{i,j}\}_{j=1}^{M}$, where $N$ and $M$ are the fixed number of context and target datapoints respectively. The context data is used to perform an update to the model $f$ such that the type of update differs depending on the model. In Neural Processes, these updates refer to its conditioning step where embeddings of the dataset are computed. Afterwards, the update is evaluated on the target data and the update rule is adjusted. In this setting, we use a neural network to model the probabilistic predictive distribution $p_\theta(y|x, \mathcal{D}^{\text{context}})$ where $\theta$ are the parameters of the NN.

### 2.2 NEURAL PROCESSES

Neural Processes (NPs) are a class of models that define an infinite family of conditional distributions where one can condition on arbitrary number of context datapoints (labelled datapoints) and make predictions for an arbitrary number of target datapoints (unlabelled datapoints), while preserving invariance in the ordering of the contexts. Several NP models have proposed to model it as follows:

$$p(y|x, \mathcal{D}_{\text{context}}) := p(y|x, r_C) \tag{1}$$

with $r_C := \text{Agg}(\mathcal{D}_{\text{context}})$ where Agg is a deterministic function varying across the variants that aggeregates $\mathcal{D}_{\text{context}}$ into a finite representation with permutation invariance in the context dataset. These NPs aim to maximise to likelihood of the target dataset given the context dataset. Conditional Neural Processes (CNPs) (Garnelo et al., 2018a), proposes an aggregator using DEEPSETs (Zaheer et al., 2017) and Conditional Attentive Neural Processes (CANPs) (Kim et al., 2019) (a deterministic variant of ANPs without the latent path) proposes an aggeragator using a single self-attention layer. In these NP variants, for a new task, we first (in a conditioning step) compute the embedding of the context dataset $r_C$. Afterwards, in a separate querying step, the model makes predictions for target datapoints. Note that for a given context dataset and independent of the number of batches of predictions, the conditioning step needs only be performed once.

Neural Processes have several desirable properties (1) **Scalability**: NPs generate predictions more computationally efficiently than Gaussian Processes which scale cubically with the size of the context dataset. In Table 1, we show the complexity of NP variants. It suffices to perform the conditioning step once but for each prediction, the querying step needs to be performed independently. As such, the complexity of the query step is the most important when considering the model's efficiency. (2) **Flexibility**: NPs can condition on an arbitrary number of context datapoints and make predictions on an arbitrary number of target datapoints, and (3) **Permutation invariance**: the predictions are order invariant in the context datapoints.

### 2.2.1 TRANSFORMER NEURAL PROCESSES

Transformer Neural Processes (TNPs) (Nguyen & Grover, 2022) comprises of a transformer-like architecture that takes both the context dataset and the target dataset as input. However, unlike previous NP variants that model the predictive distribution of target datapoints independently, i.e., $p(y_i|x_i, \mathcal{D}_{\text{context}})$, TNPs propose to model the predictive distribution of all target datapoints in conjunction $p(y_{1,...,M}|x_{1,...,M}, D_{\text{context}})$ as follows:

$$p(y_{1,...,M}|x_{1,...,M}, D_{\text{context}}) := p(y_{1,...,M}|r_{C,T}) \tag{2}$$

with $r_{C,T} := \text{Agg}(x_C, y_C, x_T)$ where Agg comprises of multiple Self-Attention layers with a masking mechanism such that the context datapoints and target datapoints only attend to the context datapoints. The output of the aggregator $r_{C,T}$ that corresponds to the target datapoints embeddings are passed to a predictor model for prediction.

Three variants of TNPs have been proposed: TNP-D, TNP-ND, and TNP-A. However, TNP-A, as the authors highlighted has low tractability due to being an autoregressive model. As such, TNP-D and TNP-ND are proposed as the practical methods for comparison. Specifically, TNP-D proposes a predictor model comprising of a vanilla MLP which outputs the mean and a diagonal covariance matrix for the target datapoints. In contrast, TNP-ND outputs a full covariance matrix for all target datapoints. TNP-ND's predictor architecture comprises of self-attention layers applied to the embeddings of target datapoints and a network to output a low-rank matrix to compute a full covariance matrix either via cholesky decomposition or a low-rank decomposition. We highlight that although the ND predictor architecture was only introduced for TNP, in fact, it can be easily extended to other NP variants such as CNP, CANP, and BNP. In which case, it would result in models such as CNP-ND, CANP-ND, and BNP-ND.

In contrast to previous NP variants, TNPs computes its embeddings with both the context dataset and target dataset. Thus, there is no conditioning step for TNPs. Instead, when making queries for a batch of target datapoints, the self-attention on both context and target datapoints must be re-computed, requiring quadratic computation $O((N+M)^2)$ in both the number of context datapoints $N$ and number of target datapoints $M$. The quadratic complexity makes deployment of the model (1) computationally costly for deployment and (2) restricts the model from scaling to settings with larger amounts of context or target datapoints.

## 3 METHODOLOGY

In this section, we introduce Latent Bottlenecked Attentive Neural Processes (LBANPs), an NP variant that achieves competitive results while being computationally efficient. Firstly, we show how to fix the quadratic computation query issue in TNPs. Secondly, we propose, LBANP, a model that utilises a latent bottlenecked attention mechanism to reduce the querying step's complexity to

constant computation per target datapoint and conditioning step's overhead to linear computation. Lastly, we show how LBANP can be extended to other variants similar to TNP depending on the problem setting.

### 3.1 REDUCING QUERYING COMPLEXITY

Due to passing both the context and query dataset through a self-attention mechanism, for TNP, each prediction requires a quadratic amount of computation in both $N$ and $M$. Unlike previous attentive-based NP variants (Kim et al., 2019), the multiple self-attention layers in EQTNP allow for higher order interactions between the target and context datapoints.

This way, we can perform cross-attention to retrieve information from the context dataset for predictions. Unlike in TNP whose queries require quadratic computation $O((N + M)^2)$ in both the number of context and target datapoints, the queries can instead be made in $O(NM)$. Figure 1 describes how the architecture of TNP is modified to give a more efficient variant that we name Efficient Queries TNPs (EQTNPs). Although EQTNPs are more efficient than TNPs, they unfortunately still require quadratic computation to compute the embeddings of the context dataset, limiting its applications.

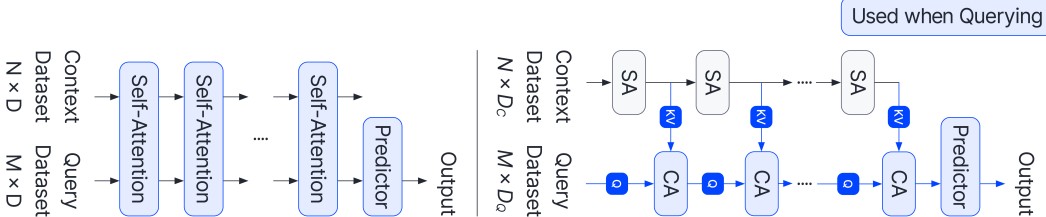

Figure 1: Model architecture for TNPs (left) and EQTNPs (right). The blue highlights the parts of the architecture used when querying. SA refers to Self-Attention and CA refers to CrossAttention.

### 3.2 LATENT BOTTLENECKED ATTENTIVE NEURAL PROCESSES

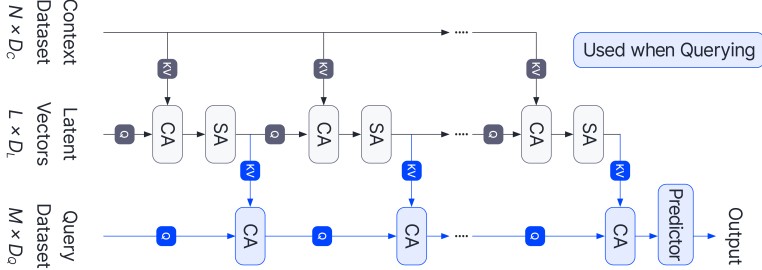

Figure 2: Architecture for Latent Bottlenecked Attentive Neural Processes (LBANPs). The blue highlights the parts of the architecture used when querying.

Inspired by the recent successes in efficient attention mechanisms (Lee et al., 2019; Jaegle et al., 2021b), we propose Latent Bottlenecked Attentive Neural Processes (LBANPs), an NP variant that reduces the quadratic conditioning complexity to linear and further reduces the query complexity to linear in the number of target datapoints, while retaining similar performance. See Figure 2 for the architecture.

**Conditioning Phase:** Instead of performing Self-Attention on the embeddings of all datapoints, we propose to reduce the dimensionality of the context dataset embeddings to a set of $L \times D_L$ latent vectors LEMB where $\text{LEMB}_0$ is meta-learned during training. The size of the latent vectors $(L \times D_L)$ represents the size of the latent bottleneck and is set according to prespecified hyperparameters $L$ and $D_L$. By interleaving retrieving information from the context dataset (CONTEXT) via a cross-attention and performing self-attention, we can compute embeddings of the context dataset that

comprise of higher-order interactions between datapoints that is typically achieved by a multilayer self-attention mechanism. Note that LBANPs also satisfy the property of being invariant to the ordering of the context and target dataset. By scaling the number of latent vectors, we can increase the size of the bottleneck, allowing for more expressive models.

$$\text{LEMB}_i = \text{SelfAttention}(\text{CrossAttention}(\text{LEMB}_{i-1}, \text{CONTEXT}))$$

Since the cross-attention is between the latent vectors and context dataset, thus the cost of this computation is $O(NL)$. Furthermore, since the self-attention is only performed on a set of $L$ latent vectors, thus it only requires $O(L^2)$ computation. Therefore, the amount of computation required to compute these latent vectors is linear in the number of context datapoints i.e., $O(NL + L^2) = O(N)$ since $L$ is a prespecified constant. The outputs of the conditioning phase are the latent embeddings $\text{LEMB}_i$ representative of the context dataset. In contrast with EQTNP, the number of (latent) vectors representing the context dataset is a constant and is thus independent of the number of context datapoints. This crucial difference contributes to a significant improvement in the overall computational complexity over both EQTNP and TNP.

**Querying Phase:** The querying phase is similar to that of EQTNP in that multiple cross-attention layers are used to retrieve information from the context dataset. The multiple cross-attention layers allow for the model to compute higher-order information between a target datapoint and the context datapoints to use in its prediction. Specifically, this is performed as follows:

$$\text{QEMB}_0 = \text{QUERY}$$
$$\text{QEMB}_i = \text{CrossAttention}(\text{QEMB}_{i-1}, \text{LEMB}_i)$$
$$\text{Output} = \text{Predictor}(\text{QEMB}_\text{K})$$

Notice that computing QEMB (short for Query Embeddings) is dependent on the number of latent vectors (a constant $L$) and is independent of the number of context datapoints. As a result, making the prediction of a single datapoint is overall constant in computational complexity, i.e., $O(1)$. Naturally, for a batch of $M$ datapoints, the amount of computation thus would be $O(M)$. Furthermore, note that LEMB only requires a constant amount of memory to store regardless of the number of datapoints. As such, unlike TNP, only constant memory is required to make a query per datapoint, allowing the method to be more applicable in settings with limited-resources.

### 3.2.1 LBANP VARIANTS

Similar to TNP-Not Diagonal (TNP-ND) (Nguyen & Grover, 2022) which predicts a covariance matrix for batch of targets, we can propose a similar variant of LBANP for settings where the predictions on target datapoints ought to be conditionally dependent. When making predictions using the target datapoint embeddings, TNP-ND first applies a self-attention mechanism solely on the target datapoint embeddings before passing into an MLP that outputs a low-rank matrix that is used to construct approximations of the full covariance matrix e.g., via Cholesky decomposition. Similarly, we could follow the same framework, resulting in a method called LBANP-ND.

## 4 EXPERIMENTS

We evaluate Latent Bottlenecked Attentive Neural Processes (LBANPs) on several tasks: meta regression, image completion, and contextual bandits. These experiment settings have been used extensively to benchmark NP models in prior works (Garnelo et al., 2018a; Kim et al., 2019; Lee et al., 2020; Nguyen & Grover, 2022). We compare LBANPs with the following members of the NP family: Conditional Neural Processes (CNPs) (Garnelo et al., 2018a), Neural Processes (NPs) (Garnelo et al., 2018b), Bootstrapping Neural Processes (BNPs) (Lee et al., 2020), and Transformer Neural Processes (TNPs) (Nguyen & Grover, 2022). In addition, we compare with their attentive variants (Kim et al., 2019) (CANPs, ANPs, and BANPs). Implementation details of the baselines and LBANPs are included in the Appendix [1]. For the sake of consistency, we evaluated LBANPs with a fixed number of latent vector ($L = 8$ or $L = 128$).

In this experiments section, we focus on comparisons with the well-established vanilla versions of NP models. Specifically, for the case of TNPs, we directly compare with TNP-D. Due to its masking

---

[1]The code is available at https://github.com/BorealisAI/latent-bottlenecked-anp.

mechanism and architecture, TNP-D models the probabilistic predictive distribution identical to that of previous NP variants. As such, it offers a fair comparison to prior NP works. In contrast, TNP-ND and TNP-A model the predictive distribution differently. TNP-A in particular has low tractability due to its autoregressive nature. For a fair comparison of TNP-ND with prior works, it would require, modifying the existing NP methods into ND variants (e.g., CNP-ND, CANP-ND, and BNP-ND). However, this is outside the scope of our work. Nonetheless, for the sake of completeness, we (1) include results for TNP-ND and TNP-A in the tables and (2) detailed how to construct the ND variants previously in Section 3.2.1 and include comparisons with TNP-ND in the Appendix.

The aim of the experimental evaluation is to answer the following questions: (1) What is the performance of LBANP compared to the existing NP methods? (2) Does LBANP scale to problems with larger number of context datapoints? (3) How does the number of latent vectors affect the performance of LBANPs?

## 4.1 IMAGE COMPLETION

In this problem, the model observes a subset of pixel values of an image and predicts the remaining pixels of the image. As a result, the problem can be perceived as a 2-D meta-regression problem in which $x$ is the coordinates of a pixel and $y$ is the value of the pixel. The image can be interpreted as a unique function in itself (Garnelo et al., 2018b). For these experiments, the $x$ values are rescaled to [-1, 1] and the $y$ values are rescaled to $[-0.5, 0.5]$. Randomly selected subsets of pixels are selected as context datapoints and target datapoints. For these experiments, we consider two datasets: EMNIST (Cohen et al., 2017) and CelebA (Liu et al., 2015).

EMNIST comprises of black and white images of handwritten letters of $32 \times 32$ resolution. In total, 10 classes are used for training, $N \sim \mathcal{U}[3, 197)$ context datapoints are sampled, and $M \sim \mathcal{U}[3, 200 - N)$ target datapoints are sampled. CelebA comprises of colored images of celebrity faces. We evaluate the models on settings of various resolutions. For CelebA32, the images are down-sampled to $32 \times 32$ and $N \sim \mathcal{U}[3, 197)$ and $M \sim \mathcal{U}[3, 200 - N)$. For CelebA64, the images are down-sampled to $64 \times 64$ and $N \sim \mathcal{U}[3, 797)$ and $M \sim \mathcal{U}[3, 800 - N)$. For CelebA128, the images are down-sampled to $128 \times 128$ and $N \sim \mathcal{U}[3, 1597)$ and $M \sim \mathcal{U}[3, 1600 - N)$.

**Results:** As shown in Table 3), using only 8 latent vectors, LBANP with only 8 latent vectors significantly outperforms all baselines except for TNP-D. In addition, we see in both CelebA and EMNIST experiments that going from 8 to 128 latent vectors improves the performance. For CelebA $(32 \times 32)$, using 128 latent vectors, LBANP outperforms TNP-D, achieving state-of-the-art results. For CelebA $(64 \times 64)$, a few attention-based NP variants (ANP and BANP) were not able to be trained due to the quadratic memory required during training. This issue was also previously reported by Garnelo et al. (2018a). Furthermore, we show that by increasing the number of latent vectors (see Section 3.2.1 and the Appendix) LBANP can perform competitively with TNP-D. For CelebA $(128 \times 128)$, due to its large number of datapoints, all existing attention-based NPs (CANP, ANP, BANP, and TNP-D) were not able to be trained. Our results show that LBANP outperforms all trainable baselines (CNP, NP, and BNP) by a significant margin. This demonstrates the advantage of our proposed method in terms of scalability and performance. Let $(N + M)_{max}$ represent the maximum number of total points (including both context and target points). In the CelebA $(128 \times 128)$ task, $L = 8 = 0.5\%(N + M)_{max}$ and $L = 128 = 8\%(N + M)_{max}$.

## 4.2 1-D REGRESSION

In this experiment, the model observes a set of $N$ context points pertaining from an unknown function $f$. The model is expected to make predictions for a set of $M$ target datapoints that comes from the same function. In each training epoch, $B = 16$ functions are sampled from a GP prior with an RBF kernel $f_i \sim GP(m, k)$ where $m(x) = 0$ and $k(x, x') = \sigma_f^2 \exp\left(-\frac{(x-x')^2}{2l^2}\right)$. The hyperparameters $l$ and $\sigma_f$ are randomly sampled per task. The hyperparameters are sampled as follows: $l \sim \mathcal{U}[0.6, 1.0)$, $\sigma_f \sim \mathcal{U}[0.1, 1.0)$, $N \sim \mathcal{U}[3, 47)$, and $M \sim \mathcal{U}[3, 50 - N)$.

At test-time, the models are evaluated on unseen functions sampled from GPs with RBF and Matern $5/2$ kernels. Specifically, the methods are evaluated according to the log-likelihood of the targets. The number of context datapoints $N$ and the number of evaluation points $M$ are sampled according to the same distribution as in training.

| Method | CelebA | | | EMNIST | |
|---|---|---|---|---|---|
| | 32x32 | 64x64 | 128x128 | Seen (0-9) | Unseen (10-46) |
| CNP | 2.15 ± 0.01 | 2.43 ± 0.00 | 2.55 ± 0.02 | 0.73 ± 0.00 | 0.49 ± 0.01 |
| CANP | 2.66 ± 0.01 | 3.15 ± 0.00 | — | 0.94 ± 0.01 | 0.82 ± 0.01 |
| NP | 2.48 ± 0.02 | 2.60 ± 0.01 | 2.67 ± 0.01 | 0.79 ± 0.01 | 0.59 ± 0.01 |
| ANP | 2.90 ± 0.00 | — | — | 0.98 ± 0.00 | 0.89 ± 0.00 |
| BNP | 2.76 ± 0.01 | 2.97 ± 0.00 | — | 0.88 ± 0.01 | 0.73 ± 0.01 |
| BANP | 3.09 ± 0.00 | — | — | 1.01 ± 0.00 | 0.94 ± 0.00 |
| TNP-D | 3.89 ± 0.01 | 5.41 ± 0.01 | — | 1.46 ± 0.01 | 1.31 ± 0.00 |
| EQTNP | 3.91 ± 0.10 | 5.29 ± 0.02 | — | 1.44 ± 0.00 | 1.27 ± 0.00 |
| LBANP (8) | 3.50 ± 0.05 | 4.39 ± 0.02 | 4.94 ± 0.03 | 1.34 ± 0.01 | 1.05 ± 0.01 |
| LBANP (128) | 3.97 ± 0.02 | 5.09 ± 0.02 | 5.84 ± 0.01 | 1.39 ± 0.01 | 1.17 ± 0.01 |
| TNP-ND | 5.48 ± 0.02 | — | — | 1.50 ± 0.00 | 1.31 ± 0.00 |
| TNP-A | 5.82 ± 0.01 | — | — | 1.54 ± 0.01 | 1.41 ± 0.01 |

Table 3: Image Completion Experiments. Each method is evaluated with 5 different seeds according to the log-likelihood (higher is better). The "dash" represents methods that could not be run because of the large memory or time requirement.

**Results:** As shown in Table 2, with 8 latent vectors, LBANP outperforms all the baselines by a large margin except TNP-D. Moreover, when using 128 latent vectors instead of 8, LBANP has competitive performance with TNP-D. It is important to note that the performance of LBANP can be further improved by increasing the number of latent vectors as shown in Figure 3a. In this task, $L = 8 = 16\%(N + M)_{max}$ and $L = 128 = 256\%(N + M)_{max}$

| Method | RBF | Matern 5/2 |
|---|---|---|
| CNP | 0.26 ± 0.02 | 0.04 ± 0.02 |
| CANP | 0.79 ± 0.00 | 0.62 ± 0.00 |
| NP | 0.27 ± 0.01 | 0.07 ± 0.01 |
| ANP | 0.81 ± 0.00 | 0.63 ± 0.00 |
| BNP | 0.38 ± 0.02 | 0.18 ± 0.02 |
| BANP | 0.82 ± 0.01 | 0.66 ± 0.00 |
| TNP-D | 1.39 ± 0.00 | 0.95 ± 0.01 |
| EQTNP | 1.32 ± 0.01 | 0.92 ± 0.01 |
| LBANP (8) | 1.20 ± 0.02 | 0.75 ± 0.02 |
| LBANP (128) | 1.27 ± 0.02 | 0.85 ± 0.02 |
| TNP-ND | 1.46 ± 0.00 | 1.02 ± 0.00 |
| TNP-A | 1.63 ± 0.00 | 1.21 ± 0.00 |

Table 2: 1-D Meta-Regression Experiments with log-likelihood metric (higher is better). All experiments are run with 5 seeds.

### 4.3 CONTEXTUAL BANDITS

In this problem previously introduced in Riquelme et al. (2018), a unit circle is divided into 5 regions: 1 low reward region and 4 high reward regions. A scalar $\delta$ determines the size of the low reward region. The 4 high reward regions have equal sizes. The agent, however, has to select amongst 5 arms (each representing one of the regions) given its 2-D co-ordinate $X$ and its actions in previous rounds. If $||X|| < \delta$, then the agent is within the low reward region. In this scenario, the optimal action is to select arm 1 which provides a reward $r \sim \mathcal{N}(1.2, 0.012)$. The other arms provide a reward $r \sim \mathcal{N}(1.0, 0.012)$. If the agent is in a high-reward region (i.e., $||X|| > \delta$), then the optimal arm would be to pull one of the other arms corresponding to the region. If the optimal arm is pulled, then the agent receives a reward of $N \sim \mathcal{N}(50.0, 0.012)$. Pulling arm 1 will reward the agent with $r \sim \mathcal{N}(1.2, 0.012)$. Pulling any of the other 3 arms rewards the agent with $r \sim \mathcal{N}(1.0, 0.012)$.

In each training iteration, $B = 8$ different wheel problems are sampled $\{\delta_i\}_{i=1}^{B}$ according to a uniform distribution $\delta \sim \mathcal{U}(0, 1)$. For the iteration, $M = 50$ points are sampled for evaluation and $N = 512$ points are sampled as context. Each datapoint is a tuple $(X, r)$ where $X$ is the co-ordinate and $r$ is the reward values for the 5 arms. The training objective is to predict the reward values for all 5 arms given the co-ordinates. We evaluate LBANP and EQTNP on settings with varying $\delta$ values. In each of the experiments, we report the mean and standard deviation of the cumulative regret for 50 different seeds for each value of $\delta$ where each run corresponds to 2000 steps. At each step, the agent selects the arm according to its Upper-Confidence Bound (UCB) and receives the reward value from the arm. The performance is measured according to the cumulative regret.

**Results:** Table 4 show that LBANPs outperform all methods except for TNPs by a significant margin. For lower $\delta$ values, we see that LBANP performs similarly to TNP-D and occasionally outperforms TNP-D in the case of $\delta = 0.95$ even with 8 latent vectors. Interestingly, we see that EQTNP

| Method | $\delta = 0.7$ | $\delta = 0.9$ | $\delta = 0.95$ | $\delta = 0.99$ | $\delta = 0.995$ |
|---|---|---|---|---|---|
| Uniform | $100.00 \pm 1.18$ | $100.00 \pm 3.03$ | $100.00 \pm 4.16$ | $100.00 \pm 7.52$ | $100.00 \pm 8.11$ |
| CNP | $4.08 \pm 0.29$ | $8.14 \pm 0.33$ | $8.01 \pm 0.40$ | $26.78 \pm 0.85$ | $38.25 \pm 1.01$ |
| CANP | $8.08 \pm 9.93$ | $11.69 \pm 11.96$ | $24.49 \pm 13.25$ | $47.33 \pm 20.49$ | $49.59 \pm 17.87$ |
| NP | $1.56 \pm 0.13$ | $2.96 \pm 0.28$ | $4.24 \pm 0.22$ | $18.00 \pm 0.42$ | $25.53 \pm 0.18$ |
| ANP | $1.62 \pm 0.16$ | $4.05 \pm 0.31$ | $5.39 \pm 0.50$ | $19.57 \pm 0.67$ | $27.65 \pm 0.95$ |
| BNP | $62.51 \pm 1.07$ | $57.49 \pm 2.13$ | $58.22 \pm 2.27$ | $58.91 \pm 3.77$ | $62.50 \pm 4.85$ |
| BANP | $4.23 \pm 16.58$ | $12.42 \pm 29.58$ | $31.10 \pm 36.10$ | $52.59 \pm 18.11$ | $49.55 \pm 14.52$ |
| TNP-D | $1.18 \pm 0.94$ | $1.70 \pm 0.41$ | $2.55 \pm 0.43$ | $3.57 \pm 1.22$ | $4.68 \pm 1.09$ |
| EQTNP | $0.75 \pm 0.15$ | $1.37 \pm 0.22$ | $2.04 \pm 0.09$ | $7.87 \pm 0.45$ | $11.15 \pm 0.13$ |
| LBANP (8) | $1.28 \pm 0.22$ | $1.83 \pm 0.24$ | $2.09 \pm 0.16$ | $7.43 \pm 0.47$ | $10.41 \pm 0.20$ |
| LBANP (128) | $1.11 \pm 0.36$ | $1.75 \pm 0.22$ | $1.65 \pm 0.23$ | $6.13 \pm 0.44$ | $8.76 \pm 0.15$ |
| TNP-ND | $1.76 \pm 0.61$ | $1.41 \pm 0.98$ | $1.61 \pm 1.65$ | $4.98 \pm 2.84$ | $7.22 \pm 3.28$ |
| TNP-A | $3.67 \pm 4.88$ | $4.04 \pm 2.38$ | $4.29 \pm 2.36$ | $5.79 \pm 5.27$ | $9.29 \pm 7.62$ |

Table 4: Contextual Multi-Armed Bandit Experiments with varying $\delta$. Models are evaluated according to cumulative regret (lower is better). Each model is run 50 times for each value of $\delta$.

outperforms TNP-D substantially for $\delta = 0.7, 0.9, 0.95$. In this task, $L = 8 = 0.4\%(N + M)_{max}$ and $L = 128 = 6.4\%(N + M)_{max}$

## 4.4 ABLATION STUDIES

### 4.4.1 NUMBER OF LATENTS

In this experiment (see Figure 3a) on CelebA ($32 \times 32$), we vary the number of latent vectors from 8 until 256 to evaluate the effect of different number of latent vectors. The experiments show that by scaling the number of latent vectors, the performance increases. This is inline with our previous results that show that LBANP with 128 latent vectors outperforms LBANP with 8 latent vectors. Specifically, in this CelebA32 setting, we see that 128 latent vectors already achieves state-of-the-art results but results can be further improved with 256 latent vectors. Furthermore, we can see that even with 8 latent vectors LBANPs achieve strong results.

### 4.4.2 EMPIRICAL ANALYSIS OF COMPUTATIONAL COMPLEXITY

**Empirical Time and Memory Cost:** In these experiments, we show the empirical time cost to perform predictions with various Neural Process models. In Figure 3b and 3c, we fix the number of target datapoints and vary the number of context datapoints. We see that TNPs required memory and time scales quadratically. In contrast, we see that EQTNPs and LBANPs remain efficient regardless of the number of datapoints. Specifically for LBANPs, it can be observed that LBANP queries utilize the same amount of time and memory as the number of context datapoints increases. These results confirm the big-O complexity analysis in Table 1. As an example, when $N = 1400$, LBANP ($L = 128$) uses 1172 MBs of memory and TNP uses 13730 MBs of memory (more than 11 times the memory LBANP used), showing a significant benefit of using LBANPs in practice.

## 5 RELATED WORK

The first NP model was Conditional Neural Processes (CNPs) (Garnelo et al., 2018a). CNPs encode the context dataset by encoding the context datapoints via a deep sets encoder. NP (Garnelo et al., 2018b) proposes to use a global latent variable to encode functional stochasticity. Later, other variants proposed to build in translational equivariance (Gordon et al., 2019), attention to tackle underfitting (Kim et al., 2019), transformer attention (Nguyen & Grover, 2022), bootstrapping (Lee et al., 2020), modeling predictive uncertainty (Bruinsma et al., 2020), and tackling sequence data (Singh et al., 2019; Willi et al., 2019). Concurrent with our work, Rastogi et al. (2023) also proposed an attention-based NP variant with linear complexity based on inducing points. From a practical perspective, NPs have also been applied to a large range of problems, including modeling sequences of stochastic processes (Singh et al., 2019), sequential decision making settings (Galashov et al., 2019), and modeling stochastic physics fields (Holderrieth et al., 2021). Mathieu et al. (2021) propose

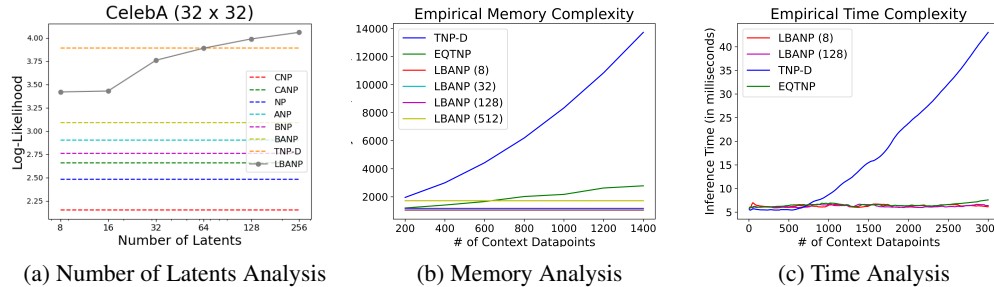

(a) Number of Latents Analysis      (b) Memory Analysis      (c) Time Analysis

Figure 3: (a) Analyzing the relation between number of latent vectors in LBANP and the model performance. Since the other models do not have a latent bottleneck, their performance is consistent and are portrayed as a dotted line. (b) Comparison of TNPs, EQTNPs, and LBANPs in terms of the amount of memory to make a prediction. (c) Comparison of TNPs, EQTNPs, and LBANPs in terms of the amount of time to make a prediction with respect to the number of context datapoints.

a contrastive learning framework to learn better representations for NPs. For an in-depth survey, we refer the reader to the recent survey paper on Neural Processes (Jha et al., 2022).

Transformers have been shown to achieve amazing results while being very flexible, but the quadratic scaling in terms of the number of inputs due to its self-attention layers renders it inapplicable to settings with large amounts of inputs e.g., an image. Several groups have proposed strategies in image settings to reduce the size of the input to the Transformer e.g., subsampling the input (Chen et al., 2020) and preprocessing the input using convolutions (Dosovitskiy et al., 2020; Wu et al., 2020). Other groups have proposed to modify the internal self-attention mechanism of Transformers to obtain enhanced efficiency. Specifically, Set Transformers and related works (Lee et al., 2019; Goyal et al., 2021) induce a latent bottleneck by mapping an input array back and forth via cross-attention between an array with fewer elements or manipulating the size of the array Jaegle et al. (2021b;a). Perceiver (Jaegle et al., 2021b) proposes a deep latent network which takes as input a simple 2D byte array (e.g., pixels) and evaluates the proposed model on image classification, audio, video, and point clouds. Perceiver IO (Jaegle et al., 2021a) extends Perceiver to inputs and outputs of different sizes.

## 6 CONCLUSION

Prior NP methods have generally been split between (1) computationally efficient (sub-quadratic) variants of Neural Processes but poor overall performance and (2) computationally expensive (quadratic) attention-based variants of Neural Processes that perform well. In this work, we fix the slow querying of the previous state-of-the-art method, TNPs, by proposing to apply self-attention only on the context dataset and retrieve information to the target datapoints via cross-attention mechanisms. Taking the idea a step further, we introduce Latent Bottlenecked Neural Processes (LBANPs) a new member of the NP family that achieves results comparable with the state-of-the-art while being computationally efficient. Specifically, the deployed LBANPs have a linear cost in the number of context datapoints for conditioning on the context dataset. Afterwards, each query requires linear computation in the number of target datapoints. In our experimental results, we show that LBANPs achieve results comparable to state-of-the-art while being more scalable. In our analysis, we show that LBANPs can trade-off the computational cost and performance according to a single hyperparameter ($L$: the number of latent vectors). This allows a user to easily select the expressivity of LBANPs depending on the computational requirement of the setting. Importantly, unlike with TNPs, the amount of computation (both time and memory) required to query a single target datapoint for LBANPs will always be constant regardless of the number of context datapoints.

## REPRODUCIBILITY STATEMENT

The code is available at https://github.com/BorealisAI/latent-bottlenecked-anp. Hyperparameters and implementation details are available in the Appendix, Section 3, and Section 4.

ACKNOWLEDGEMENTS

The authors acknowledge funding from CIFAR and the Quebec government.

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

## A  APPENDIX: ADDITIONAL METHODOLOGY DETAILS

### A.1  EFFICIENT QUERY TRANSFORMER NEURAL PROCESSES

**Conditioning Phase:** In the conditioning phase, we compute the embeddings representative of the context dataset

$$\mathrm{CEMB}_0 = \mathrm{CONTEXT}$$
$$\mathrm{CEMB}_i = \mathrm{SelfAttention}(\mathrm{CEMB}_{i-1})$$

Since the number of context datapoints is $O(N)$, the computational cost per SelfAttention layer is $O(N^2)$. Let $K$ be the number of self-attention layers. Then the embeddings $\{\mathrm{CEMB}_1, \ldots, \mathrm{CEMB}_K\}$ encode the model's knowledge of the context dataset.

**Querying Phase:** When making predictions, we make predictions by retrieving information from the context dataset, specifically from $\{\mathrm{CEMB}_1, \ldots, \mathrm{CEMB}_K\}$, via multiple cross-attention layers.

$$\mathrm{QEMB}_0 = \mathrm{QUERY}$$
$$\mathrm{QEMB}_i = \mathrm{CrossAttention}(\mathrm{QEMB}_{i-1}, \mathrm{CEMB}_i)$$
$$\mathrm{Output} = \mathrm{Predictor}(\mathrm{QEMB}_{\mathrm{K}})$$

Each CrossAttention layer is linear in CEMB. Since there is an embedding per datapoint, the computational cost is $O(NM)$. The multiple cross-attention layers that retrieve information from the context dataset embeddings allows the predictor to retrieve higher-order information just like in TNPs.

### A.2  COMPUTATIONAL AND MEMORY COMPLEXITY

In Table 5, we include the computational and memory complexity for the NP variants with $N$ context datapoints and $M$ target datapoints. Note that in the case of a complexity of $O(M)$, the computational and memory cost for a single target datapoint is constant.

| Method | Computational (and Memory) Complexity (Big-O) | | |
| | Training | Evaluation | |
| | Step | Condition | Query |
| --- | --- | --- | --- |
| CNP | $N + M$ | $N$ | $M$ |
| CANP | $N^2 + NM$ | $N^2$ | $NM$ |
| NP | $N + M$ | $N$ | $M$ |
| ANP | $N^2 + NM$ | $N^2$ | $NM$ |
| BNP | $N + M$ | $N$ | $M$ |
| BANP | $N^2 + NM$ | $N^2$ | $NM$ |
| TNP-D | $(N + M)^2$ | — | $(N + M)^2$ |
| **EQTNP** | $N^2 + NM$ | $N^2$ | $NM$ |
| **LBANP** | $(N + M + L)L$ | $NL + L^2$ | $ML$ |
| TNP-ND | $(N + M)^2$ | $N^2$ | $(N + M)^2$ |
| **LBANP-ND** | $N + M^2$ | $N$ | $M^2$ |
| **LBANP-END** | $N + M^2$ | $N$ | $M^2$ |

Table 5: Computational and Memory Complexity in Big-O notation of the model with respect to the number of context datapoints $N$ and number of target datapoints in a batch $M$. $L$ and $K$ are prespecified hyperparameters. $L$ is the number of latent vectors. $K$ is the number of bootstrapping samples for BNP and BANP. When using a NP at test-time, when making multiple predictions with the model, conditioning is a one-time cost but querying is not. As such, it is most important that the cost of querying is low.

## B APPENDIX: ADDITIONAL EXPERIMENTAL DETAILS AND RESULTS

### B.1 IMPLEMENTATION AND HYPERPARAMETER DETAILS

We use the implementation of the baselines from the official repository of TNPs (see https://github.com/tung-nd/TNP-pytorch and its respective paper (Nguyen & Grover, 2022)). The hyperparameters for the CelebA experiments are the same regardless of the resolution. EQTNP experiments follow almost identically the TNP experiment hyperparameters. The only difference is that EQTNPs architecture is slightly different than that of TNP. Specifically, EQTNP has a cross-attention mechanism to retrieve information from the context dataset. The number of cross-attention layers in EQTNP is equivalent to that of the number of self-attention layers in TNP. Similarly, LBANP hyperparameters are almost identical to that of EQTNP and that of TNP. LBANP consists of blocks each comprising of a cross-attention mechanism from the context dataset to the latent space and a self-attention mechanism in the latent space. In LBANP, the number of blocks for the conditioning phase is equivalent to the number self-attention mechanisms in the conditioning phase of EQTNP.

Due to its cross-attention mechanisms, LBANP allows for varying embedding sizes for the context dataset $D_C$, latent vectors $D_L$, and query dataset $D_C$. For simplicity, we set $D_C = D_L = D_Q = 64$, following TNP's embedding size of $64$. Similarly for EQTNP, we set $D_c = D_Q = 64$. We do not tune the number of latent vectors ($L$). Instead, we showed results for LBANP with $L = 8$ and $L = 128$ latent vectors. The remainder of the hyperparameters is the same for LBANP, EQTNP, and TNP.

For the ND experiments, we follow TNP-ND (Nguyen & Grover, 2022) and use cholesky decomposition for our LBANP-NP and LBANP-ENP experiments. All experiments were either run on a GTX 1080ti (12 GB RAM) or P100 GPU (16 GB RAM). When verifying if computationally expensive models were trainable for CelebA64 and CelebA128, we used the P100 GPU (the GPU with larger amounts of RAM).

### B.2 ANALYSIS: NUMBER OF LATENTS

In this section, we include an additional plot (see Figure 4) that analyses the effect of varying the number of latent vectors. We clearly see that scaling the number of latent vectors improves the performance of LBANP. At 512 latent vectors, we see that LBANP achieves performance competitive with that of TNP-D.

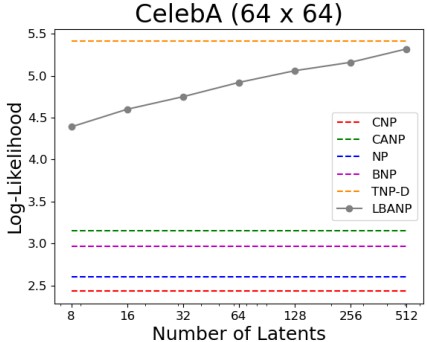

Figure 4: Analysis regarding the number of latent vectors in LBANP.

### B.3 ABLATION: INTERLEAVING CROSS-ATTENTION LAYERS

Naively an idea is to first compute embeddings for the context dataset $N$ and pass it to a TransformerDecoder like layer, but this incurs a quadratic complexity in the number of target datapoints. Alternatively, retrieving information via a single cross-attention layer. However, this limits the amount of information that can be retrieved which affects performance. We show that in this ablation. Specifically, we verify whether interleaving the cross-attention layers is important for LBANP. In this experiment, we consider LBANP-L (LBANP-Last) where instead of multiple cross-attention layers

for querying, the model only uses a single cross-attention layer which attends over the last latent embedding. In Table 6, we see that LBANP outperforms LBANP-L. As such, indeed interleaving the multiple cross-attention layers to retrieve higher-order information from the context dataset for the target datapoint is important.

| Method | CelebA | |
| --- | --- | --- |
| | 32x32 | 64x64 |
| LBANP-L (8) | $3.35 \pm 0.01$ | $4.10 \pm 0.04$ |
| LBANP (8) | $3.50 \pm 0.05$ | $4.39 \pm 0.02$ |
| LBANP-L (128) | $3.78 \pm 0.03$ | $4.90 \pm 0.05$ |
| LBANP (128) | $3.97 \pm 0.02$ | $5.09 \pm 0.02$ |

Table 6: Ablation Image Completion Experiments.

### B.4 ANALYSIS: COMPUTATIONAL COMPLEXITY

#### B.4.1 EMPIRICAL TIME COMPLEXITY

In these experiments, we show the empirical time cost to perform predictions with various Neural Process models. In Figure 5a, we fix the number of target datapoints and vary the number of context datapoints. In Figure 5b, we fix the number of context datapoints and vary the number of target datapoints.

In both experiments, we see that TNP's require significantly more time as the number of datapoints scale. In contrast, we see that EQTNPs and LBANPs remain efficient regardless of the number of datapoints. Importantly, we see that regardless of the number of context datapoints, LBANPs queries use the same amount of time.

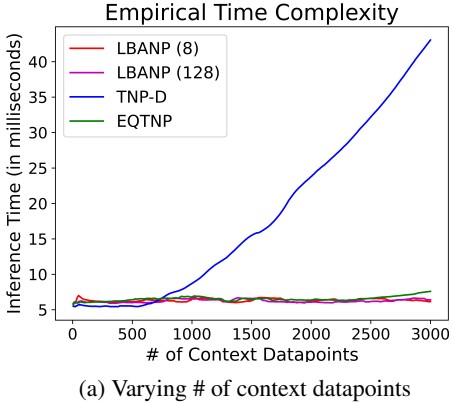

(a) Varying # of context datapoints

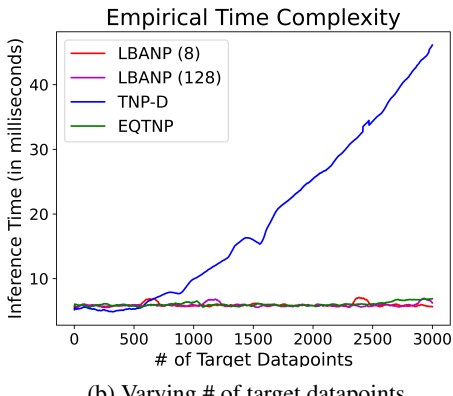

(b) Varying # of target datapoints

Figure 5: Comparison of TNPs, EQTNPs, and LBANPs in terms of the amount of time to make a prediction.

#### B.4.2 EMPIRICAL MEMORY COMPLEXITY

In these experiments, we show the empirical memory cost to perform predictions with various Neural Process models. In Figure 6a, we fix the number of target datapoints and vary the number of context datapoints. The figure shows that with respect to the number of context datapoints: (1) TNP's memory cost scales quadratically, (2) EQTNP's memory cost scales linearly, and (3) LBANP's memory cost stays constant. These results confirm the big-O complexity analysis in Table 1. When $N = 1400$, LBANP ($L = 128$) uses 1172 MBs of memory and TNP uses 13730 MBs of memory (more than 11 times the memory LBANP used), showing a significant benefit of using LBANPs in practice.

In Figure 6b, we fix the number of context and target datapoints and vary only the number of latent vectors for LBANPs. The plot shows that the memory cost of LBANPs indeed scales linearly with the number of latent vectors.

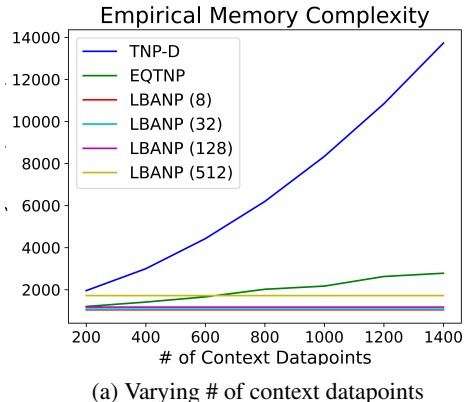

(a) Varying # of context datapoints

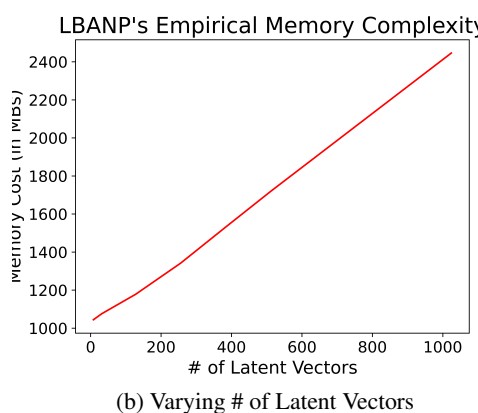

(b) Varying # of Latent Vectors

Figure 6: (a) Comparison of TNPs, EQTNPs, and LBANPs in terms of the amount of memory to make a prediction. (b) Plot of the amount of memory to make a prediction relative to the number of latent vectors.

In Figure 7, we include a scatterplot of the memory used when querying for a varying number of context datapoints ($N$) and target datapoints ($M$). The plot shows that TNP-D uses significantly more memory than LBANPs when making predictions. Importantly, we notice that TNP's memory increases significantly with gains in performance as the number of datapoints increase. In contrast, LBANP does not require significantly more memory to achieve the large performance gains. The memory used for querying is in fact constant as the number of context datapoints increase.

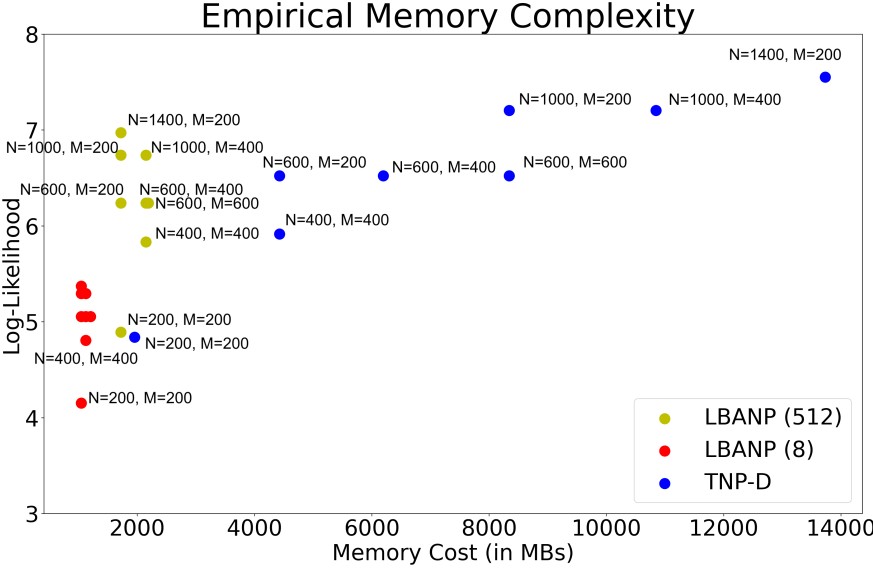

Figure 7: Scatterplot of the memory used for a varying number of context datapoints ($N$) and target datapoints ($M$). Since the performance of LBANP (8) is comparable as the number of datapoints increase. As such, for clarity, the majority of datapoints for LBANP (8) are without text specifying $N$ and $M$.

### B.4.3 ANALYSIS: CONVERGENCE

Following TNPs, we run our experiments for 200 epochs. In Fig 8, we see that TNPs converge slightly quicker than that of LBANPs and EQTNPs. By comparing LBANP ($L = 128$) and LBANP ($L = 8$), we see that increasing the number of latent vectors help LBANPs learn quicker.

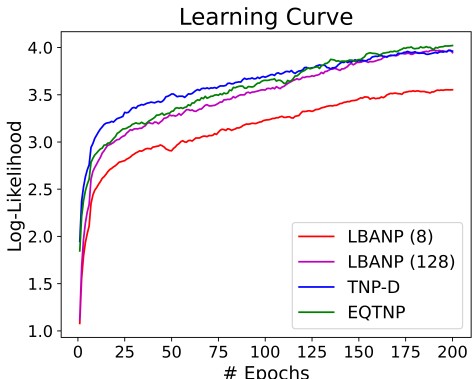

Figure 8: Learning Curve comparison of TNPs, EQTNPs, and LBANPs.

### B.5 NOT-DIAGONAL VARIANTS

In this section, we consider Not-Diagonal variants of Neural Processes. Unfortunately the self-attention mechanism used in Not-Diagonal models (such as TNP-ND or LBANP-ND) require quadratic computation with respect to the number of target datapoints when querying. As a result, although LBANP-ND is efficient in terms of the number of context datapoints, the model would not be scalable in terms of the large number of target datapoints. Instead, we propose a more efficient variant called LBANP-END (Efficient Not Diagonal). Instead of a self-attention mechanism which has a quadratic computational cost, LBANP-END proposes a latent bottlenecked attention mechanism that is linear in terms of computation. Specifically, the predictor information from the target datapoint embeddings via a cross-attention mechanism to a fixed set of latent vectors LQEMB (Latent Query Embeddings). The initialization of this set of latent vectors $\text{LQEMB}_0$ is meta-learned. We apply a self-attention mechanism to compute the higher-order information in latent space. After the higher-order information has been computed, we retrieve the information from the latent vectors via another cross-attention mechanism. Specifically, this is performed as follows:

$$\text{LQEMB}_i = \text{SelfAttention}(\text{CrossAttention}(\text{LQEMB}_{i-1}, \text{QEMB}_K))$$
$$\text{Output} = \text{Predictor}(\text{CrossAttention}(\text{QEMB}_K, \text{LQEMB}_Q))$$

where $Q$ is a hyperparameter representing the number of layers.

We evaluate the performance of the non-diagonal variants of the methods on the image completion task. Table 7 shows that LBANP-ND performs competitively with TNP-ND across both the CelebA and EMNIST datasets. In the table, we include TNP-D and LBANP (128) as a reference of the performance of the vanilla variants. In the results, we see that both TNP-ND and LBANP-ND are incapable of scaling to CelebA64. In contrast, we see that, our proposed efficient non-diagonal variant, LBANP-END is more scalable than LBANP-ND and TNP-ND, and achieves state-of-the-art results on both the $32 \times 32$ and $64 \times 64$. Finally, similar to previous experiments, we see that scaling the number of latent vectors improves the performance of these variants as well.

| Method | CelebA | | EMNIST | |
|---|---|---|---|---|
| | 32x32 | 64x64 | Seen (0-9) | Unseen (10-46) |
| TNP-D | 3.89 ± 0.01 | 5.41 ± 0.01 | 1.46 ± 0.01 | 1.31 ± 0.00 |
| LBANP (128) | 3.97 ± 0.02 | 5.09 ± 0.02 | 1.39 ± 0.01 | 1.17 ± 0.01 |
| TNP-ND | 5.48 ± 0.02 | — | 1.50 ± 0.00 | 1.31 ± 0.00 |
| LBANP-ND (8) | 5.14 ± 0.01 | — | 1.38 ± 0.01 | 1.06 ± 0.01 |
| LBANP-ND (128) | 5.57 ± 0.03 | — | 1.42 ± 0.01 | 1.14 ± 0.01 |
| LBANP-END (8) | 5.14 ± 0.02 | 5.36 ± 0.03 | 1.39 ± 0.01 | 1.06 ± 0.01 |
| LBANP-END (128) | 5.60 ± 0.02 | 6.06 ± 0.03 | 1.42 ± 0.02 | 1.14 ± 0.04 |

Table 7: Image Completion Experiments with Not Diagonal variants in terms of log-likelihood.

