# OpenReview forum: "Latent Bottlenecked Attentive Neural Processes"
_ICLR.cc/2023/Conference — ICLR 2023 poster_

### Official Review · Reviewer_peD2 · 2022-10-15

**Confidence:** 4
**Correctness:** 3
**Technical Novelty And Significance:** 2
**Empirical Novelty And Significance:** 2
**Recommendation:** 3

**Clarity, Quality, Novelty And Reproducibility:**

The paper is written in a clear and precise scientific language. I only find Figures 1 and 2 not sufficiently descriptive. They were not useful for me to understand the core idea, as they contain only a complex block diagram with lots of abbreviations. I would prefer the description of the *original idea* instead of architectural details.

The paper contains sufficient details to reproduce the reported results.

The proposed method is novel only in the sense that some well-known architectural elements such as latent embeddings and cross-attention are used in the very context probably for the first time.


--- After rebuttal ---

While I appreciate very much the authors' effort, I'm afraid my concerns are not addressed. The author response echoes the statements already mentioned in the paper, which I already read carefully and understood before writing my review. And I do not find these very arguments convincing. NPs are designed for small context sets as they are typical building blocks for few-shot learning setups. In the cases where one needs to extract context from massive data, one can always employ representative point selection (or learning) methods or simply backpropagate the gradients up till the context entries (or their low-dimensional embeddings) akin to the learnable inducing points for Gaussian Processes. While the solution presented in the paper has some degree of originality, I'm afraid I still do not see its scientific value. Furthermore, the experiments do not appear to be designed to prime realistic applications where a big context sets could indeed be a problem.

**Strength And Weaknesses:**

**Strengths:**

 * The paper reports a comprehensive set of experimental results, where the proposed method appears to indeed approach the TNP performance in reduced computational cost.
* The proposed idea is rather simple, easy to grasp and straightforward to implement. It is an original, though trivial, combination of a number of known architectural attention structures.


**Weaknesses:**

 * It is not clear to me why quadratic complexity on the context set is such a severe limitation. Is it not the common assumption of NPs that the context set is rather small? Similarly, I assume the target queries are iid when conditioned on the context. Then how critical is it in reality for a target batch to be big? Can one simply not process small query batches even in parallel?
* The proposed novelty is very incremental and straightforward. It does indeed improve performance, but I would categorize the presented method rather as a useful implementation trick than as a solid scientific finding.


**Summary Of The Paper:**

The paper studies the computational bottleneck problem of Transformer Neural Processes (TNPs) that results from the quadratic complexity of transformers with respect to the input sequence length. The paper aims to combine the best of the worlds of state-of-the-art expressive power at the expense of high compute cost (TNPs) and scalable computation at the expense of suboptimal performance (other NP variants). The paper proposes a modification to the TNP pipeline to trade these competing goals using a fixed-sized attentive memory. The main idea is to embed the input into a fixed-dimensional latent space and model high-order interactions among data points as cross-attention to this embedding as opposed to self attention in the native observation space.

**Summary Of The Review:**

While being a decent piece of work with some degree of originality and some promising results, I find the scientific contribution of the paper too slim for a main-track ICLR paper. I am also not yet convinced about the significance of the studied problem (see first bullet item under Weaknesses).

---

> ### Author Response · Authors · 2022-11-14
> **Response to Reviewer peD2 (1/2)**
>
>
> > It is not clear to me why quadratic complexity on the context set is such a severe limitation. Is it not the common assumption of NPs that the context set is rather small?
>
>
> Neural Processes are general-purpose meta-learned uncertainty estimators that crucially do not require retraining on new tasks, allowing for a broad range of applications. They are particularly useful in settings with limited compute that are unable to train their own uncertainty estimators.
>
> However, even in scenarios without strong limitations on the compute, scalable NPs are very useful. One example is in multi-round scenarios such as Contextual Bandits or Bayesian Optimization. These settings train uncertainty estimators such as ensembles or MC-Dropout on potentially large initial training data (e.g. thousands of samples or more). Furthermore, these uncertainty models require retraining the model each round where a new sample is added to the training set. In total, the number of rounds can potentially be large, resulting in the retraining being prohibitively expensive both in time and compute. As such, it is highly beneficial to have good uncertainty estimators that are both scalable and cheap to update with new data. Neural Processes offer such a benefit.
>
>
> Furthermore, the case that the context set is small is also due to the limited applicability of previous existing NP works. In general, existing NP methods are not scalable. For example, in Table 3, besides LBANPs, only NPs and CNPs were able to run on CelebA (128 x 128). However, they perform very poorly, only achieving  2.55 and 2.67 log-likelihood respectively. In the same setting, LBANP (L=128) achieved 5.84 log-likelihood, more than doubling the log-likelihood of CNPs and NPs. The existing well-performing NP variants such as TNPs are simply not able to be applied to such settings due to their computational complexity.
>
>
>
> > Similarly, I assume the target queries are iid when conditioned on the context. Then how critical is it in reality for a target batch to be big?
>
>
> Whether or not target queries are iid when conditioned on the context depends on the task. For example, in the popular image completion experiments (e.g., CelebA and EMNIST), the target queries (i.e., the missing pixels) are not iid. In such non-iid settings, being able to predict a large target batch is beneficial (hence the very strong performance of LBANP-ND, LBANP-END, and TNP-ND). In settings such as Bayesian Optimization, however, the target queries are treated independently. In such settings, it is not as critical to handle large target batch sizes; nonetheless, being able to make predictions for large target batches will certainly save overall compute.
>
> > Can one simply not process small query batches even in parallel?
>
> In the case that the target datapoints are non-iid, then splitting the target datapoints into smaller batches is detrimental both to predictive performance and computational cost.
>
> In the case that the target datapoints are iid, then as you mentioned, we could process the different query batches in parallel. However, in that scenario, LBANPs would be even more computationally superior to the previous state-of-the-art TNPs.
>
> For example, consider a scenario where we would want to process $100$ batches of size $M=100$ with a context dataset of size $N=1600$. If all the batches were done in parallel, then TNP would require $100$ computations of $O((N+M)^2) = O((1600 + 100)^2)$. Due to a large amount of computation, it is likely that it would require the memory of several GPUs. In contrast, LBANP would only require $100$ computations of $O(M) = O(100)$, requiring thus only a single GPU.

---

> > ### Author Response · Authors · 2022-11-14
> > **Response to Reviewer peD2 (2/2)**
> >
> >
> > > The proposed novelty is very incremental and straightforward.
> > > It does indeed improve performance, but I would categorize the presented method rather as a useful implementation trick than as a solid scientific finding.
> >
> > We believe this work is practically important to the community since the current state-of-the-art transformer-based architecture is intractable for a large number of datapoints. However, the new proposed architecture (LBANP) can scale to the large problem settings as shown in the paper (results for CelebA 128x128, Figures 3(b)-3(c), and Appendix B.4.1). So, this paper can be classified under new architecture discovery papers with useful outcomes such as performance improvement and complexity management.
> >
> > We encourage the reviewer to see Appendix B.4.1 and B.4.2, where we perform an analysis on the computational complexity of the proposed model in terms of time and memory. It can be observed that the new model architecture provides significant improvement over TNPs. In fact, TNPs time cost grows substantially with respect to both $N$ and $M$, resembling a quadratic with a low coefficient (likely due to the efficient parallelisation of computation performed by GPUs). In contrast, EQTNPs and LBANPs are significantly more efficient. Importantly, it can be observed that LBANPs time complexity remain constant regardless of the value of $N$.
> >
> >
> > > While being a decent piece of work with some degree of originality and some promising results, I find the scientific contribution of the paper too slim for a main-track ICLR paper. I am also not yet convinced about the significance of the studied problem (see first bullet item under Weaknesses).
> >
> >
> > Our contributions are as follows:
> >  - We show that TNPs are inefficient due to the architectural design. As such, we propose to compute embeddings in a conditioning phase and to make predictions via retrieving in multiple cross-attention layers, denoting this method as EQTNPs.
> >  - Building on EQTNPs, we introduce LBANPs a method which uses a latent bottlenecked attention mechanism, allowing for representing the context dataset in a fixed number of latent vectors. This allows LBANPs to (1) be significantly cheaper in deployment while retaining similar performance to TNPs (2) trade-off computational cost and performance according to a hyperparameter and (3) scale NPs to harder settings, allowing for broader applicability of NPs

---

> ### Author Response · Authors · 2022-11-19
> **A Gentle Reminder to Reviewer peD2**
>
> Dear Reviewer peD2,
>
> We have addressed all of your concerns in the comment below. In particular, we have highlighted the importance of the problem setting (i.e., designing scalable Neural Processes). We have detailed how target queries are not necessarily iid. In the case of non-iid, it is crucially important to scale to be able to scale to a large number of target queries. Furthermore, we explain how in the small query batch size scenario that you've described, the computational advantages of using LBANPs grow significantly.
>
> In our results, we show that all previous attention-based NP models are not tractable for large problem settings. Our results and analyses show that (1) LBANPs can scale to problems beyond existing attention-based NPs, while outperforming all the computationally-efficient NP variants by a large margin (2) LBANPs are significantly more efficient than TNPs in terms of both time and memory. Specifically, it can be seen that LBANPs computational cost remains constant regardless of the number of context data points. Due to the clear benefits of LBANPs, we believe this work is practically important and useful to the community.
>
> Could you please let us know if there are other reasons you think this work is below the acceptance threshold?
> Please let us know if you have any further questions. We are happy to address any questions or concerns you have. Any feedback would be highly appreciated.
> We look forward to hearing from you.

---

> ### Author Response · Authors · 2022-11-27
> **Kind Reminder to Reviewer peD2**
>
> We have addressed your concerns in the rebuttal:
>
>  - (1) **NP context sizes are small:** We detailed how this is due to the limited applicability (scalability) of current NP variants. This limited applicability of NP variants is in fact exactly what our proposed method is aiming to solve.
>  - (2) **Target queries are iid when conditioned on the context:** We explained how this is not true for many settings, including the popular image completion settings for Neural Processes.
>  - (3) **Can one simply not process small query batches even in parallel?** We detailed how considering small query batches in parallel further improves our method compared to SOTA baselines.
>
> We further showed our proposed method provides very significant benefits compared to the previous state-of-the-art in terms of scalability, memory, and time while retaining similar performance.
>
> Since we have addressed your concerns, are there any other specific reasons you think this work is below the acceptance threshold? We would highly appreciate any feedback. We are happy to address any other concerns you have.

---

> ### Author Response · Authors · 2022-12-03
> **Reply to Reviewer peD2**
>
> We thank the reviewer for their response.
>
> > I'm afraid I still do not see its scientific value
>
> Unlike typical learning-based methods, NPs crucially does not require any special training regime (e.g., backpropagation ) when given the context dataset.
> Being able to simply deploy a pre-trained model for ready uncertainty estimation given a dataset is already a **significant** advantage.
> This advantage is shared by a recent work TabPFN (Hollmann et al., 2022). TabPFN is a meta-learned transformer-based model such that "[g]iven a new data set, there is no costly gradient-based training. Rather, it's a single forward pass of a fixed network" (Hutter, 2022)
> In our work, we consider the regression setting and have scaled to settings with **larger** datasets than that of TabPFN.
>
> > NPs are designed for small context sets as they are typical building blocks for few-shot learning setups.
>
> NPs have also been applied to settings with larger context sets in previous works. For example, Contextual Bandit experiments previously used in TNPs (Nguyen et al., 2022) had up to 1,000 context datapoints. CelebA (64x64), previously used in ConvCNPs (Gordon et al., 2019), also does not have a small context set.
>
> > In the cases where one needs to extract context from massive data, one can always employ representative point selection (or learning) methods or simply backpropagate the gradients up till the context entries (or their low-dimensional embeddings) akin to the learnable inducing points for Gaussian Processes.
>
> As mentioned in the previous comment, not having to manually learn and train is a considerable advantage of our proposed method over methods such as learnable inducing points for GPs. Furthermore, learning the inducing points itself is **quadratic**(!) in the number of inducing points $O(nm^2)$ (where $n$ is the total number of datapoints and $m$ is the number of inducing points), limiting the scalability to harder settings.
>
>
> ---
>
> F. Hutter [@FrankRHutter]. "This may revolutionize data science...". Twitter, 21 Oct 2021, https://twitter.com/FrankRHutter/status/1583410845307977733
>
> N. Hollmann, S. Müller, K. Eggensperger, F. Hutter. "TabPFN: A Transformer That Solves Small Tabular Classification Problems in a Second". arXiv:2207.01848, 2022
>
> T. Nguyen, A. Grover. "Transformer Neural Processes: Uncertainty-Aware Meta Learning Via Sequence Modeling". In International Conference on Machine Learning, 2022
>
> J. Gordon, W. P. Bruinsma, A. Y. K. Foong, J. Requeima, Y. Dubois, R. E. Turner. "Convolutional Conditional Neural Processes". In International Conference on Learning Representations, 2020

---

> ### Author Response · Authors · 2022-12-06
> **Reminder to Reviewer peD2**
>
> Dear Reviewer peD2,
>
> We are sending a reminder since the discussion period will be ending in less than a week. We hope that our previous response addressed your concerns in your edited review ("---After rebuttal---").
>
> We would like to highlight that LBANPs offer more than just scalability to large context sizes and performance comparable to state-of-the-art performance.
> 1. Importantly, LBANPs can trade-off computational cost (both in terms of memory and time) and performance (according to the number of latent vectors). This flexibility is **unique to LBANP** and is non-existent in existing NP variants, including TNPs which is the state-of-the-art.
> 2. LBANPs scale to large number of target datapoints which is beneficial for many settings. For example,
>     **(i)** settings that evaluate large number of target points such as Bayesian Optimization when optimizing the acquisition function.
>     **(ii)** settings with not-iid targets when conditioned on the context. Due to the non-iid nature of the targets, large batch predictions allow for better uncertainty estimates in settings such as the popular image completion tasks in the paper.
>
> We are more than happy to address any further concerns that you may have. We look forward to hearing from you.

---

> ### Author Response · Authors · 2022-12-09
> **Reminder to Reviewer peD2**
>
> Dear Reviewer peD2,
>
> We are sending you a reminder that the discussion period will be ending in $3$ days. We have addressed your concerns in the below comments, including highlighting the benefits of LBANPs beyond scaling to large context sizes. We are more than happy to address any further concerns that you may have. We look forward to hearing from you.

---

> ### Author Response · Authors · 2022-12-12
> **NP Survey: Cost-effective generalisation is a major issue faced by Neural Processes**
>
> Dear Reviewer peD2,
>
> We would like to bring your attention to the recent survey paper on Neural Processes. In Section 7 on "Future Research Directions", "cost-effective generalisation" is highlighted as the first major issue. Specifically, the quadratic computational cost is highlighted as an issue of current NPs that should be fixed. Furthermore, Gordon et al., 2020 also highlights the issue of limited scalability of attention-based NP variants.
>
> This "cost-effective generalisation" issue of current state-of-the-art NPs is precisely the problem which our proposed method LBANPs tackles. We hope this clarifies the scientific value of our work. Once again, we are more than happy to address any further concerns that you may have.
>
> ---
>
> Jonathan Gordon, Wessel P. Bruinsma, Andrew Y. K. Foong, James Requeima, Yann Dubois, and Richard E.
> Turner. Convolutional conditional neural processes. In International Conference on Learning Representations, 2020.

---

### Official Review · Reviewer_vZAK · 2022-10-23

**Confidence:** 4
**Correctness:** 3
**Technical Novelty And Significance:** 2
**Empirical Novelty And Significance:** 2
**Recommendation:** 6

**Clarity, Quality, Novelty And Reproducibility:**

The paper is well organized and written,  and figures and tables are clear and easily to understand. It is easy to follow the motivation and idea.

However, the novelty is a little weak. The only contribution is to introduce the iterative attention to replace the previous cross-attention in transformer NP without much analysis on the possible effects.

**Strength And Weaknesses:**

Strength

1. The motivation is clear and the proposed idea looks reasonable.
2. There are extensive empirical evaluations and the performance looks good and comparable with transformer NP

Weaknesses

1. Considering the main contribution of this work is to reduce the computational cost, there should be some empirical evaluation on the time complexity in addition to table 6 so let the readers know how much time we can save from transformer NP when achieves similar results.

2. The table 6 is a little unfair, because there is L in the new iterative attention. As shown in Figure 4, the size of L apparently heavily affects on the final performance. It would be better to include it in the time complexity.

3. It is expected to see more discussions or analysis on the effect of this new attention comparing with previous one. For example, will it affect on the convergence rate? Will it break the process property of the neural processes because the proposed method introduce more `constraints' on the latent variables which are assumed to be independent before, like NP and CNP? like permutation invariant and marginal consistency?




**Summary Of The Paper:**

This paper proposes a new member of neural processes. The aim is to improve the attention efficiency of transformer neural process where the attention strategy requires quadratic computation with respect to the number of context data points.  The idea is from iterative attention (Jaegle et al., 2021b) which changes the self-attentions between context data points to a cross-attention with latent variable and a self-attention between latent variable. When the size of the latent variable is significantly smaller than the one of context size, such an exchange could greatly reduce the total computational cost.

**Summary Of The Review:**

The paper is well written and easy to follow. The motivation is clear and method is reasonable and effective. The main weakness is the in-depth analysis on the new iterative attention to the base neural process. please see the `Strength And Weaknesses' for more details.


---- after rebuttal

Thanks to the authors to clarify my concerns in the response. I do believe the proposed method could save a significant amount of computational time and memory according to the new results. Some other issues are resolved as well. I appreciate that. So I am happy to change my score from 5 to 6. The reason why I did not give 8 is the novelty reason, although I believe the proposed technique could improve the efficiency of TNP a lot and I recognize its value to the area, the technique itself is mainly from the existing work - iterative attention. I read the authors' response (to all reviewers) carefully but there is no explanation of the 'new' design and technique proposed compared with iterative attention.

---

> ### Author Response · Authors · 2022-11-14
> **Response to Reviewer vZAK**
>
>
> We thank the reviewer for their helpful feedback.
>
> > Considering the main contribution of this work is to reduce the computational cost, there should be some empirical evaluation on the time complexity in addition to table 6.
>
> Thank you for your suggestion!
>
> Please see appendix B.4.1, where we've included plots of the time consumption (comparing TNPD, EQTNP, LBANP ($L = 8$ and $L = 128$)) at test time of querying as a function of $N$ (the number of context datapoints) and $M$ (the number of target datapoints). We see that TNPs time cost grows substantially with respect to both $N$ and $M$, resembling a quadratic with a low coefficient (likely due to the efficient parallelisation of computation performed by GPUs). In contrast, EQTNPs and LBANPs ($L=8$ and $L=128$) are significantly more efficient. Importantly, we see that LBANPs remain constant both for $L=8$ and $L=128$ regardless of the size of $N$.
>
>
>
> > The table 6 is a little unfair, because there is L in the new iterative attention. As shown in Figure 4, the size of L apparently heavily affects on the final performance. It would be better to include it in the time complexity.
>
> Thank you for your suggestion.
>
> To reduce confusion since BNP and BANP also introduce a hyperparameter that affects the complexity, we made the table with only the number of context ($N$) and target datapoints ($M$). We update the table to include the hyperparameters of the different NP variants.
>
>
> > will it affect on the convergence rate?
>
>
> In Appendix B.4.3, we have compared the convergence of TNPs, EQTNPs, and LBANPs. Following the TNP experimental setup for CelebA (32x32), we run our experiments for 200 epochs. It can be seen that all methods continue to learn for all epochs and converge at approximately the same rate. In addition by comparing LBANP ($L=128$) and LBANP ($L = 8$), we see that increasing the number of latent vectors helps LBANP learn in the earlier epochs.
>
>
>
> > Will it break the process property of the neural processes because the proposed method introduce more constraints on the latent variables which are assumed to be independent before, like NP and CNP?
> > like permutation invariant and marginal consistency?
>
> This is a great question. Both permutation invariance and marginal consistency are preserved in LBANPs.
>
> Similar to that of ANPs, the permutation invariance is preserved since a cross-attention mechanism retrieves information via keys/values from the context dataset and cross-attention mechanisms are order invariant in the keys/values, i.e., order invariant in the context datapoints.
>
> During the conditioning phase, the latent vectors are learned so they can summarize the whole information of the context dataset for making predictions. Similar to ANPs, the cross-attention mechanism preserves its marginal consistency since cross-attention treats each query independently. Since LBANPs only use cross-attention mechanisms to retrieve the information for the targets, therefore LBANPs also preserve marginal consistency.
>
> However, for the Not-Diagonal variants (LBANP-ND), similar to TNP-ND, the marginal consistency is lost due to Self-Attention being applied to the embeddings of the targets.
>
> Side note: a similar argument shows that EQTNPs also preserve permutation invariance and marginal consistency.
>
>
>
> > It is expected to see more discussions or analysis on the effect of this new attention comparing with previous one.
>
> Thank you for your valuable suggestions. As discussed in the previous point,  we have explained how LBANPs uphold properties of marginal consistency and permutation invariance. In addition, we have included in the appendix an empirical analysis of the computational complexity in terms of time and memory consumption (Sections B.4.1 and B.4.2). Further, an analysis on convergence rate has been included in Appendix B.4.3.
>
> > However, the novelty is a little weak. The only contribution is to introduce the iterative attention to replace the previous cross-attention in transformer NP without much analysis on the possible effects.
>
>
> We would like to clarify that Transformer NP uses only self-attention for both the context and target datapoints.
>
> Our novelty is as follows:
>  - We show that TNPs are inefficient due to the architectural design. As such, we propose a modification which computes embeddings in a conditioning phase and makes predictions via retrieving in multiple cross-attention layers, denoting this method as EQTNPs.
>  - Building on EQTNPs, we introduce LBANPs a method which uses a latent bottlenecked attention mechanism, allowing for representing the context dataset in a fixed number of latent vectors (which are learned during training). This allows LBANPs to (1) be significantly cheaper in deployment while retaining similar performance to TNPs, (2) trade-off computational cost and performance according to a single hyperparameter, and (3) scale to harder problem settings.

---

> ### Author Response · Authors · 2022-11-19
> **A Gentle Reminder to Reviewer vZAK**
>
> Dear Reviewer vZAK,
>
> We have addressed all of your concerns in the comments below.
> As you suggested, we have revised the big-O complexity table in the paper to include the hyperparameters of Neural Process variants which affect the computational complexity.
> Furthermore, we have included analyses of the empirical memory and time usage. The analysis shows our proposed LBANPs are significantly more efficient in both time and memory than transformer NPs.
> Finally, we have also detailed how the process properties (permutation invariance and marginal consistency) are preserved and included an analysis on the convergence rate.
>
> Could you please let us know if there are specific reasons you think this work is below the acceptance threshold?
> We are happy to address any further concerns you have. Any feedback would be highly appreciated. We look forward to hearing from you.

---

> ### Author Response · Authors · 2022-11-27
> **Kind Reminder to Reviewer vZAK**
>
> We have addressed your concerns in the rebuttal:
>
>  - (1) **Lack of empirical evaluation on the time:** We have included an empirical evaluation on the time **and memory**, showing significant benefits for LBANP over prior SOTA.
>  - (2) **Include $L$ in the time complexity:** We have included $L$ in the time complexity.
>  - (3) **The effect on convergence, permutation invariance, and marginal consistency:** We detailed how LBANP does not break permutation invariance and marginal consistency. We also showed that it converges at a similar rate to prior SOTA.
>
> Since we have addressed your concerns, are there any other specific reasons you think this work is below the acceptance threshold? We would highly appreciate any feedback. We are happy to address any other concerns you have.

---

> ### Author Response · Authors · 2022-12-03
> **Reminder to Reviewer vZAK**
>
> Dear Reviewer vZAK,
>
> We are sending you a reminder that the discussion period will be ending this month. Once again, we are happy to address any further concerns that you may have. We look forward to hearing from you.

---

> ### Author Response · Authors · 2022-12-07
> **Response to Reviewer vZAK**
>
> We would like to thank the reviewer for their very constructive feedback and for their support.
>
> We would like to highlight that LBANPs offer more than efficiency compared to TNPs.
>  - (1) Importantly, LBANPs can trade-off computational cost (both in terms of memory and time) and performance (according to the number of latent vectors). This flexibility is **unique to LBANP** and is non-existent in existing NP variants, including TNPs which is the state-of-the-art.
>  - (2) LBANPs scale to settings with large context sizes. This allows for LBANPs to be applied in a wider range of settings beyond current attention-based Neural Processes.
>  - (3) LBANPs scale to large number of target datapoints which is beneficial for many settings. For example, (i) settings that evaluate large number of target points such as Bayesian Optimization when optimizing the acquisition function. (ii) settings with not-iid targets when conditioned on the context. Due to the non-iid nature of the targets, large batch predictions allow for better uncertainty estimates in settings such as the popular image completion tasks in the paper. Hence, the very strong performance of LBANP-END.
>
> Although we agree that part of the idea was proposed in iterative attention, our work is the first to:
>  - show empirical time and memory complexity analysis with varying numbers of latents
>  - extend the idea to Neural Processes such that predictions can be made in linear time and process properties (marginal consistency and permutation invariance) are preserved. The design choice is not trivial as highlighted by Reviewer RFme's questions regarding why a standard decoder was not used and why interleaving was necessary.
>  - Going beyond iterative attention, we also introduced a non-trivial variant of LBANP (specifically: LBANP-END) (Section 3.2.1) which scales beyond TNP-ND and achieves state-of-the-art results (Table 7). On CelebA (64x64), LBANP-END achieves a log-likelihood $6.06 ± 0.03$ compared to TNP's $5.41 ± 0.01$.

---

### Official Review · Reviewer_RFme · 2022-10-24

**Confidence:** 4
**Correctness:** 3
**Technical Novelty And Significance:** 4
**Empirical Novelty And Significance:** 4
**Recommendation:** 8

**Clarity, Quality, Novelty And Reproducibility:**

The closest works seem to me to be TNP and ANP. With respect to these, to my knowledge, the model is novel. The quality and clarity is good. The authors have also promised to release the code.

**Strength And Weaknesses:**

### Pros

1. The use of latent-bottleneck in the context of TNPs is novel and interesting.
2. Shows better performance than pre-TNP baselines. Claims to show better memory complexity than TNP through a big-O analysis. Naturally, the performance is expected to be worse than TNP and this is empirically confirmed. But this performance drop can be traded-off with gains in memory saving shown via the big-O complexity.
3. The comparison with EQTNP is useful as a natural variant of TNP. The fact that EQTNP performs a bit worse than TNP-D is new knowledge.

### Weaknesses/Questions

1. $L$ is treated as a pre-specified constant. However, if the tasks are indeed too complex/difficult such that they typically require a large $N$, then users are likely to choose a proportionately large  $L$ also to handle the underlying difficulty of the tasks. In other words, the task difficulty is a confounder that affects both $N$ and $L$ simultaneously. So, it might be tricky to say that the complexity is simply $O(N+M)$ while completely omitting $L$. It might be more justified to say $O((N+M)L)$?
2. In experiments, the choice of $L$ should also be better justified given what distribution of $N$ values was used (perhaps reporting $L$ as a fraction of the mean value of $N$?). For instance, in the 1-D regression problem, $N$ takes values in $[3,47)$, and in this case, showing $L=128$ (which seems much larger than any $N$) seems to defeat the purpose of having latents?
3. A way to address the above concerns might be to show the actual GPU memory consumption at test time as a function of $N$ and $L$ and compare LBANP’s memory savings relative to TNP. I am unsure what would be the best way though and I leave it up to the authors to decide the best way. Also, what was the memory cap that prevented running some experiments in Table 3?
4. The interleaving seems to make the proposed model deviate from the standard GPT layers — something that, to me, was a big appeal of TNPs. Why is interleaving important in EQTNP and also in LBANP? Why not first do encoding of the context points with a standard transformer and then later do the prediction of target points with another standard transformer decoder? An ablation could be helpful.

### Minor Comments/Questions

1. Would be good to show a row for the computational complexity of EQTNP in Table 1 rather than only in Table 6.
2. In Section 4.3, the Normal distribution may be shown with a calligraphic $\mathcal{N}$.

**Summary Of The Paper:**

TNP uses GPT layers and shows good performance. But its memory complexity $O((N+M)^2)$ could be large if $N$ is large. The paper proposes a latent-bottleneck of $L$ vectors that mediate between the $N$ context points and the $M$ target points. By doing this, the authors are able to reduce the complexity to $O(NL + ML)$ where $L$ is the number of latent vectors and a human-specified hyperparameter.

**Summary Of The Review:**

I believe in the promise and value of introducing the latent bottleneck and I tend to believe in the reduced big-O complexity. But in experiments, reporting the actual saving in memory should go hand-in-hand with the reporting of the performance. I am currently rating the paper as 6 in the hope that the authors will be able to add more justification. I am happy to revise my rating in the light of new information.

---

> ### Author Response · Authors · 2022-11-14
> **Response to Reviewer RFme (1/2)**
>
> We would like to thank the reviewer for the detailed comments and the very helpful feedback.
>
> > $L$ is treated as a pre-specified constant. However, if the tasks are indeed too complex/difficult such that they typically require a large $N$, then users are likely to choose a proportionately large also to handle the underlying difficulty of the tasks. In other words, the task difficulty is a confounder that affects both $N$ and $L$ simultaneously. So, it might be tricky to say that the complexity is simply $O(N+M)$ while completely omitting $L$. It might be more justified to say $O((N+M)L)$?
>
> You are right that the complexity includes the factor of $L$. The complexity would be $O(NL + L^2)$ for the Conditioning phase and $O(ML)$ to perform a query. The $L^2$ factor comes from the self-attention applied to the latent vectors. However, since typically $L$ is typically upperbounded by the maximum value of $N$, the complexity could potentially be reduced to $O(NL)$ for the Conditioning phase. The $NL$ and $ML$ factors are due to the cross-attention mechanism.
>
> To reduce confusion since BNP and BANP also introduce a hyperparameter that affects the complexity, we made the table with only the number of context ($N$) and target datapoints ($M$). We updated the table to include the hyperparameters of the different NP variants.
>
> > In experiments, the choice of $L$ should also be better justified given what distribution of $N$ values was used (perhaps reporting $L$ as a fraction of the mean value of $N$?). For instance, in the 1-D regression problem, $N$ takes values in $[3, 47)$, and in this case, showing $L = 128$ (which seems much larger than any $N$) seems to defeat the purpose of having latents?
>
> Thank you for the suggestion.
> We agree with you that understanding the relation between $N$ and $L$ is important for the reader to better understand the benefit of the method. For simplicity and consistency across the experiments, we used the same number of latent vectors ($L=8$ and $L=128$) across all experiments. In the results section, for each dataset, we will report $L$ as a fraction of the maximum value of $N$.
>
> > A way to address the above concerns might be to show the actual GPU memory consumption at test time as a function of N and L and compare LBANP’s memory savings relative to TNP. I am unsure what would be the best way though and I leave it up to the authors to decide the best way.
>
> Thank you for your suggestion. In Figure 3(b), we included a plot showing the empirical GPU memory as a function of $N$. Inside that we  also show the GPU memory consumption for different size of latent vector (8, 32, 128, 512). Further, we have dedicated Appendix B.4.2 and Figure 3 (b) (main paper) to the empirical GPU complexity. It can be observed the actual GPU consumption is significantly more efficient than TNPs. LBANPs GPU memory usage is constant and TNPs GPU memory scales similarly to that of a quadratic, confirming our big-O analysis. In Appendix B.4.2, we include a plot showing the empirical GPU memory as a function of $L$.
>
> In the meantime, we've also included plots of the time consumption (comparing TNPD, EQTNP, LBANP ($L = 8$), and LBANP ($L = 128$)) in Appendix B.4.1 and Figure 3 (c) (main paper) at test time of querying as a function of $N$ (the number of context datapoints) and $M$ (the number of target datapoints). We see that TNPs time cost grows substantially with respect to both $N$ and $M$, resembling a quadratic with a low coefficient (likely due to the efficient parallelisation of computation performed by GPUs). In contrast, EQTNPs and LBANPs ($L=8$ and $L=128$) are significantly more efficient. Importantly, we see that LBANPs remain constant both for $L=8$ and $L=128$ regardless of the size of $N$.

---

> > ### Author Response · Authors · 2022-11-14
> > **Response to Reviewer RFme (2/2)**
> >
> >
> > > Also, what was the memory cap that prevented running some experiments in Table 3?
> >
> > Our experiments were run on GTX 1080 Ti with a memory cap of 11GB.
> >
> > > The interleaving seems to make the proposed model deviate from the standard GPT layers — something that, to me, was a big appeal of TNPs. Why is interleaving important in EQTNP and also in LBANP? Why not first do encoding of the context points with a standard transformer and then later do the prediction of target points with another standard transformer decoder? An ablation could be helpful.
> >
> > This is a great question. We agree that using the standard GPT layers is appealing. However, the standard Transformer Decoder utilises self-attention. As such, if we were to apply a standard transformer decoder, then it would induce a quadratic complexity in the querying phase, defeating the purpose of LBANPs and EQTNPs.
> >
> > An alternative idea which retains the complexity would be to first perform the encoding and then retrieve the information for decoding via a cross-attention layer afterwards instead of interleaving. We refer to this method as LBANP-L. This is an ablation included in the Appendix that we performed. In our experiments shown, we found that interleaving empirically improved the performance, i.e., LBANP > LBANP-L.
> >
> > | Method\Resolution |      32x32      |      64x64      |
> > |:-----------------:|:---------------:|:---------------:|
> > |    LBANP-L (8)    |   3.35 ± 0.01   |   4.10 ± 0.04   |
> > |     LBANP (8)     | **3.50 ± 0.05** | **4.39 ± 0.02** |
> > |   LBANP-L (128)   |   3.78 ± 0.03   |   4.90 ± 0.05   |
> > |    LBANP (128)    | **3.97 ± 0.02** | **5.09 ± 0.02** |
> >
> > ### Minor Comments/Questions
> >
> > > Would be good to show a row for the computational complexity of EQTNP in Table 1 rather than only in Table 6.
> > > In Section 4.3, the Normal distribution may be shown with a calligraphic $\mathcal{N}$.
> >
> > Thank you for your suggestions! We've updated the paper accordingly.

---

> > > ### Author Response · Authors · 2022-11-15
> > > **Included in Appendix: GPU memory consumption plots**
> > >
> > > As requested, we have included in the Appendix plots showing the GPU memory consumption as a function of $N$ and $L$. For this analysis experiment, we used specifically an "NVIDIA Quadro RTX 5000" GPU with 16GBs of memory.
> > >
> > > **GPU memory as a function of $N$ (the number of context datapoints):**
> > >
> > > The plot (Figure 3(b)) shows: (1) TNP's memory cost scales quadratically, (2) EQTNP's memory cost scales linearly, and (3) LBANP's memory cost stays constant. These empirical results confirm the big-O complexity analysis in the paper. When $N=1400$, LBANP ($L=128$) used $1172$ MBs of memory and TNP used $13730$ MBs of memory (more than 11 times the memory LBANP used), showing a significant benefit of using LBANP in practice.
> > >
> > >
> > > **GPU memory for LBANP as a function of $L$ (the number of latent vectors):**
> > >
> > > The plot (Figure 6(b)) shows that the memory cost of LBANPs scales linearly with the number of latent vectors.

---

> > > > ### Comment · Reviewer_RFme · 2022-12-03
> > > > **Thank You**
> > > >
> > > > I sincerely thank the authors for making the effort to fix my concerns. If I could revise my rating from 6 to 7, I would have happily done it. But I can only choose between 6 and 8. To give an 8, I still feel that my main concern about clearly showing the trade-off between performance and memory is somewhat unaddressed. I was hoping to see a clear trade-off plot between memory and performance and how LBANP does better on it (e.g., a plot with memory usage on the $x$ axis and performance on the $y$ axis for varying number of context/target/latent points). Currently, the authors add a new memory-usage plot that is shown separately and is disconnected from the plot about the performance. Yes, it is true that memory/time would scale quadratically but I would have liked to see the performance values for those points too simultaneously. Perhaps, this can be something that can be added in a final version.
> > > >
> > > > But overall, I see the value of the proposed model and I think the paper can be accepted as is, but I will maintain my rating of 6 for now.

---

> > > > > ### Author Response · Authors · 2022-12-05
> > > > > **Reply to Reviewer RFme**
> > > > >
> > > > > Thank you for your response.
> > > > >
> > > > > We believe that the beneficial trade-off between memory/performance was clear from our plots. However, to completely address your concern, we are in the process of generating a scatter plot with the memory usage as the x-axis and performance as the y-axis for the final version to more clearly show how LBANP does better in trading-off between performance and memory. The plots' points will comprise of LBANPs and TNPs with varying numbers of contexts/targets/latents.
> > > > >
> > > > > From the memory-usage and performance plots included in the paper, the results of the scatter plot can already be inferred:
> > > > >  - The number of target datapoints will not affect the performance of either method since the predictions for several small batches of targets will be the same values as making the prediction for one large batch. However, TNPs memory usage will increase substantially. Therefore, LBANPs will clearly better trade-off the memory/performance.
> > > > >  - Increasing the number of context datapoints will improve the performance of both methods. TNPs memory usage will increase substantially and LBANPs memory usage will stay constant. Therefore, LBANPs will clearly better trade-off the memory/performance.
> > > > >  - The number of latents will affect the performance and memory of LBANPs. The memory will increase linearly with respect to the number of latents. The performance of LBANPs will increase as the number of latents increase.
> > > > >
> > > > > We hope this justifies revising your score to an 8. Please let us know if you have any other concerns or suggestions.

---

> > > > > > ### Comment · Reviewer_RFme · 2022-12-05
> > > > > > **Thank You**
> > > > > >
> > > > > > Thank you, this makes sense. I have now revised my score to 8. But I do hope to see the scatter plot in the final version that the authors have promised.

---

> > > > > > > ### Author Response · Authors · 2022-12-05
> > > > > > > **Thank You**
> > > > > > >
> > > > > > > We would like to thank the reviewer for their very constructive feedback and for their support. As promised, we are working on the scatter plot and we will include it in the final version of the paper.

---

> ### Author Response · Authors · 2022-11-19
> **A Gentle Reminder to Reviewer RFme**
>
> Dear Reviewer RFme,
>
> We would like to thank you for your detailed feedback.
> We have addressed all your concerns in the comments below.
> We have revised the big-O complexity table in the paper to include the hyperparameters of Neural Process variants which affect the computational complexity.
> As you suggested, we have included an analysis of the empirical memory usage as a function of $N$ and $L$. In addition, we have also included an analysis of the empirical time cost. Both the time and memory analyses show our proposed LBANPs are significantly more efficient than transformer NPs, confirming our big-O analyses.
> Lastly, we address your concern about the necessity of interleaving, showing that interleaving improves the performance of LBANP by a significant margin.
>
> Could you please let us know if these changes have addressed your concerns? We hope you raise your score based on these changes.
> Please let us know if you need any further clarification. We are happy to address any questions you have. Any feedback would be highly appreciated. We look forward to hearing from you.

---

### Author Response · Authors · 2022-11-18
**Rebuttal Phase Summary**

Dear Reviewers,

As the rebuttal phase is ending in a few hours, we wanted to summarize the discussion.

The reviewers’ major concerns were mostly about the actual computational complexity and the contributions.
We added new studies to show the empirical complexity of our model in terms of memory and time in Figures 3(b)-3(c) and with more detailed discussions in Appendix B.4.1 and B.4.2. It can be observed that the current state-of-the-art TNP model is very costly and its computational time grows substantially with respect to both $N$ (number of context points) and $M$ (number of target points), which resembles a quadratic function. However, our proposed EQTNPs and LBANPs are significantly more efficient in terms of both time and memory. Specifically, it can be seen that LBANPs computational cost remain constant regardless of the number of data points.
Actually, this ties to our main contribution, which is proposing a new model design with comparable SOTA results but significantly less computational cost. In fact, the prior TNP model is not tractable for large problem settings. For example, this can be seen in Table 3, where TNP and all previous attention-based NPs cannot scale to CelebA 128x128. But, our proposed LBANP can tackle the problem and outperform all the computationally-efficient baselines by a large margin. That is why we believe this work is practically important and useful to the community.

We also provided answers to all the reviewers' questions by replying to their comments and quoting each specific question.

---

### Decision · Program_Chairs · 2023-01-20

**Decision:**

Accept: poster

**Justification For Why Not Higher Score:**

Incremental in combining a known technique for scaling transformers (ie, iterative attention) to TNPs.

**Justification For Why Not Lower Score:**

Practically useful extension of TNPs for certain applications with high number of context points.

**Metareview: Summary, Strengths And Weaknesses:**

The paper proposes an extension of Transformer Neural Processes (TNP; Nguyen & Grover, ICML 2022) for larger context sizes. TNPs are SOTA for meta-learning in the NP family, but have inference cost that is quadratic in the total number of context and target points due to the use of self-attention. This paper proposes LBANP which extends the idea of iterative attention to TNPs by projecting the context points to a fixed number of latent vectors. By increasing the number of latent vectors, the model can tradeoff performance for memory/compute savings. The reviewers acknowledged the problem and the solution, from both conceptual and practical perspective. Experimental evidence was also fairly convincing in e.g., showing generalization to very high resolution image completion tasks.

I encourage the authors to also address a couple of concerns. One of the reviewers was not convinced with the motivation to scale to large context sets and the lack of baseline approaches for compressing context sets. The latter is a fair concern worth addressing. Finally, in the spirit of transparency, I also think the authors should show reference results with TNP-A. I believe the authors' statement " TNP-A... has low tractability due to being an autoregressive model" is only true with regards to the complexity w.r.t. target set size; the processing time complexity for the context set in TNP-A is the same as that of TNP-D and TNP-ND.

**Note From Pc:**

if the above contains the word "oral" or "spotlight" please see: "oral" presentation means -> notable-top-5% and "spotlight" means -> notable-top-25%. As stated in our emails, we are disassociating presentation type from AC recommendations